# Newly Initiated Statin Treatment Is Associated with Decreased Plasma Coenzyme Q10 Level After Acute ST-Elevation Myocardial Infarction

**DOI:** 10.3390/ijms26010106

**Published:** 2024-12-26

**Authors:** Erika Csengo, Hajnalka Lorincz, Eva Csosz, Andrea Guba, Bettina Karai, Judit Toth, Sara Csiha, Gyorgy Paragh, Mariann Harangi, Gergely Gyorgy Nagy

**Affiliations:** 1Centre of Cardiovascular Diseases and Internal Medicine, Borsod-Abauj-Zemplen County Central Hospital and University Teaching Hospital, Szentpéteri kapu 72-76, 3526 Miskolc, Hungary; 2Division of Metabolism, Department of Internal Medicine, Faculty of Medicine, University of Debrecen, 4032 Debrecen, Hungary; 3Proteomics Core Facility, Department of Biochemistry and Molecular Biology, Faculty of Medicine, University of Debrecen, Nagyerdei krt. 98, 4032 Debrecen, Hungary; 4Department of Laboratory Medicine, Faculty of Medicine, University of Debrecen, Nagyerdei krt. 98, 4032 Debrecen, Hungary; 5Doctoral School of Health Sciences, University of Debrecen, Nagyerdei krt. 98, 4032 Debrecen, Hungary; 6Institute of Health Studies, Faculty of Health Sciences, University of Debrecen, Kassai út 26, 4032 Debrecen, Hungary

**Keywords:** coenzyme Q10, ubiquinone, fatty acid-binding protein 3 (FABP3), ST-elevation myocardial infarction (STEMI), acute coronary syndrome (ACS), statin, myopathy, cholesterol

## Abstract

Coenzyme Q10 (CoQ10) plays a crucial role in facilitating electron transport during oxidative phosphorylation, thus contributing to cellular energy production. Statin treatment causes a decrease in CoQ10 levels in muscle tissue as well as in serum, which may contribute to the musculoskeletal side effects. Therefore, we aimed to assess the effect of newly initiated statin treatment on serum CoQ10 levels after acute ST-elevation myocardial infarction (STEMI) and the correlation of CoQ10 levels with key biomarkers of subclinical or clinically overt myopathy. In this study, we enrolled 67 non-diabetic, statin-naïve early-onset STEMI patients with preserved renal function. Plasma CoQ10 level was determined by ultra-high-performance liquid chromatography–tandem mass spectrometry (UPLC/MS-MS), while the myopathy marker serum fatty acid-binding protein 3 (FABP3) level was measured with enzyme-linked immunosorbent assay (ELISA) at hospital admission and after 3 months of statin treatment. The treatment significantly decreased the plasma CoQ10 (by 43%) and FABP3 levels (by 79%) as well as total cholesterol, low-density lipoprotein cholesterol (LDL-C), apolipoprotein B100 (ApoB100), and oxidized LDL (oxLDL) levels. The change in CoQ10 level showed significant positive correlations with the changes in total cholesterol, LDL-C, ApoB100, and oxLDL levels, while it did not correlate with the change in FABP3 level. Our results prove the CoQ10-reducing effect of statin treatment and demonstrate its lipid-lowering efficacy but contradict the role of CoQ10 reduction in statin-induced myopathy.

## 1. Introduction

Coenzyme Q10 (CoQ10) or ubiquinone is a lipid-soluble molecule localized principally in mitochondria, which can undergo oxidation and reduction cycles. Human tissues with high metabolic rates such as the heart, liver, or kidney contain substantial concentrations of CoQ10 [1]. It is a key component of complex-I [nicotinamide adenine dinucleotide hydrogen (NADH)–ubiquinone oxidoreductase] and III (ubiquinone–cytochrome c oxidoreductase) in the mitochondrial electron transport chain and plays an important role in oxidative phosphorylation and thus the energy metabolism of mammalian cells [2]. In addition, CoQ10 is an important endogenous antioxidant that protects membrane phospholipids from lipid peroxidation, reduces the levels of reactive oxygen species, and is capable of regenerating other intracellular antioxidants [3]. Moreover, ubiquinone has been implicated in anti-inflammatory pathways [4].

CoQ10 is mainly endogenously biosynthesized by the body itself, although dietary intake has a minor contribution to ubiquinone levels as well [5,6]. CoQ10 is composed of a polyisoprenoid tail and a benzoquinone ring. The former is synthesized in the cytosol via the mevalonate pathway, and it is attached to the latter within the mitochondria. A key enzyme in the mevalonate pathway is 3-hydroxy-3-methylglutaryl-CoA (HMG-CoA) reductase, also involved in the initial steps of cholesterol biosynthesis. Primary or secondary CoQ10 deficiencies were implicated in a wide range of clinical conditions, including cardiomyopathy, nephropathy, and neurological and sensory disorders [5,7]. It is quite clear, that organs with high energy requirements or excessive metabolic and oxidative stress are preferentially affected by CoQ10 depletion.

Acute coronary syndromes (ACS) including ST-elevation and non-ST-elevation acute myocardial infarction (STEMI; NSTEMI) are leading causes of mortality and morbidity worldwide, with long-term clinical and metabolic sequels following the acute event [8]. The ischemic damage caused by the occlusion of a major epicardial coronary artery and subsequent reperfusion injury occurring after revascularisation with primary percutaneous coronary angiography (pPCI) or fibrinolysis lead to the loss of viable myocardium tissue and heart failure (HF). The ensuing cardiac remodelling poses an energetic, metabolic, and hemodynamic burden on the remaining myocardial tissue to maintain minute volume and organ perfusion. Excess oxidative stress and inflammation are associated with these processes in the acute and subacute phases of myocardial infarction. In the meantime, lipid-lowering therapy (LLT) has become a cornerstone of secondary prevention in atherosclerotic cardiovascular diseases (ASCVDs), with numerous clinical trials demonstrating that lowering low-density lipoprotein cholesterol (LDL-C) can reduce cardiovascular events and mortality [9]. The higher the cardiovascular risk, the lower the LDL-C target; thus, LDL-C below 1.4 mmol/L (55 mg/dL) should be achieved in these very high-risk post-ACS patients. On top of the strict treatment goals, a European clinical consensus statement emerged, which proposed the lipid-lowering strategy of “strike early and strike strong”, with immediate administration of a dual lipid-lowering treatment for selected patients after ACS [10]. These stringent therapeutic goals can be achieved only with robust, lifelong, and high-intensity statin treatment inhibiting the HMG-CoA reductase.

Altogether, these pathophysiological processes and the statin-induced inhibition of the mevalonate pathway can potentially result in secondary CoQ10 deficiency after myocardial infarction (MI). This might have detrimental effects on recovery, global myocardial, or other organ functions. Indeed, several animal and human studies tried to address this hypothesis in the past, but the results are often inconclusive. CoQ10 levels are rarely reported or investigations are frequently based on experimental models where CoQ10 supplementation was administered to alleviate the consequences of myocardium ischemia [11,12,13,14,15,16]. It is accepted that statin treatment results in decreased CoQ10 levels and this does not seem to be associated with major adverse events or mitochondrial dysfunction, but results are based on studies where low- or moderate-intensity LLT regimens were used on stable patients [17,18,19,20]. It is still unknown whether an acute MI treated with a contemporary standard of care is associated with long-term changes in CoQ10 levels. This single-centre, prospective study was carried out to assess CoQ10 levels and key HF, atherosclerosis, and myotoxicity biomarkers in STEMI patients treated with pPCI and guideline-directed medical therapy at the time of the acute event and after a three-month-long follow-up period. Our main hypothesis was that high-intensity statin treatment of statin-naïve early-onset STEMI patients decreased serum CoQ10 levels and this secondary CoQ10 deficiency was correlated with an increased level of fatty acid-binding protein-3 (FABP3), a sensitive and specific biomarker of myopathy.

## 2. Results

Sixty-seven non-diabetic, statin-naïve, early-onset STEMI patients (64.2% males) with 48.7 ± 6.0 mean years were enrolled (Table 1). The mean BMI was 28.1 ± 5.3 kg/m^2^. Mean GRACE and APACHE-II scores were 82.9 ± 15.6 and 5.6 ± 2.3, respectively. Most subjects were smokers (80.6%) and treated with hypertension (64.2%) at admission. None of the patients were diabetic (type 1 or type 2), and only 4.5, 3, and 3 percent of subjects suffered from peripheral artery disease, coronary artery disease, and stroke, respectively. Median ischaemic time from symptom onset to reperfusion therapy was 4.0 h (0.5–40.0 h). The mean left ventricular ejection fraction was 51.3 + 13.0%, and the most frequent cause of STEMI was an occlusion of LAD or RCA (47.8 and 37.3%, respectively).

At admission, none of the STEMI patients were treated with statin, ezetimibe, fibrate and PCSK9 inhibitors. Statin treatment was initiated during hospitalization in all patients with STEMI according to international guidelines. Furthermore, ezetimibe and fibrate treatment were added to the statin therapy of some patients (two and two patients, respectively). Due to the limited reimbursement of PCSK9 inhibitors, these agents were not administered during hospitalization or during the 3-month follow-up. During the 3-month follow-up, one of our patients discontinued the statin therapy due to an allergic skin reaction. The main medications of patients at admission and at the 3-month follow-up are summarized in Table 1. The detailed course of lipid-lowering treatment regimens during the study is summarized in Appendix A.

An initially not planned, optional remote visit was carried out after a median long-term follow-up time of 36 (24.8–46.7) months. Several attempts were made to contact each patient via a documented telephone call. The great majority of patients (*n* = 59, 88%) were available for this visit, while the rest were lost to follow-up (*n* = 8, 12%). Only one male patient (1.49%) reported symptoms of muscle pain, which led to the discontinuation of statin treatment. A rechallenge was not attempted in his case; therefore, the SAMS-CI questionnaire could not be completed. However, based on the characteristics of myalgia (asymmetric, intermittent, not specific to any area), the late onset of pain (>12 weeks), and no improvement of muscle symptoms 4 weeks after statin withdrawal, it was unlikely that the patient’s muscle symptoms were caused by statin use. Three additional patients were not adherent to statin treatment, with two due to fears of side effects, and one because of imaging abnormalities found in the liver. Patients were also queried regarding their diet. None of them reported restrictive diets, which might have had a significant effect on CoQ10 intake compared to the general population.

Laboratory parameters of STEMI patients at admission and after 3-month statin treatment are shown in Table 2. Plasma CoQ10 concentration markedly decreased after 3-month statin treatment (at admission: 43.5 (28.4–52.9) vs. after 3 months: 24.66 (17.8–34.4) ng/mL; *p* = 2.0 × 10^−6^) (Table 2 and Figure 1a). Neither baseline nor 3-month CoQ10 levels were significantly different between men and women (*p* = 0.973 and *p* = 0.882, respectively). Ratios of CoQ10/total cholesterol, CoQ10/LDL-C, and CoQ10/Apo B100 were calculated. Only the CoQ10/LDL-C ratio increased significantly (Table 2).

Heart/muscle fatty acid-binding protein (FABP3) levels, which is a biomarker for cardiac or skeletal muscle injury, decreased significantly during the follow-up (at admission: 4.66 (2.21–13.53) vs. after 3 months: 0.98 (0.83–1.46) ng/mL; *p* = 2.7 × 10^−11^) (Table 2 and Figure 1b). Also, circulating serum levels of NT-proBNP, creatine kinase, CRP, and IL-6 significantly decreased.

Among lipid parameters, triglyceride, total cholesterol, LDL-C, Apo B100 and oxLDL levels significantly decreased, while Apo AI levels increased during statin treatment (Table 2 and Figure 1c–f). Serum HDL-C, γ-GTP, and creatinine levels remained unchanged. The levels of AST and ALT significantly improved and reached the normal range during the follow-up.

Baseline CoQ10 concentration did not correlate with age (r = 0.211; *p* = 0.092), while there was a positive correlation between baseline CoQ10 and BMI (r = 0.286; *p* = 0.022) in STEMI patients. To analyze the role of obesity, multiple regression analysis was performed where plasma CoQ10 was applied as a dependent variable (Appendix A). Based on this analysis, we found that total cholesterol was an independent predictor of CoQ10 (β_standardized_ = 0.34; *p* = 0.01), while BMI did not predict the plasma level of CoQ10.

During the 3-month follow-up, there were no correlations between the change in FABP3, NT-proBNP, CRP, IL-6, liver enzymes, creatinine, uric acid, triglyceride, and CoQ10 (Table 3). In turn, there were significant positive correlations between the change in total cholesterol, LDL-C, Apo B100, oxLDL, and CoQ10 levels (Figure 2). Furthermore, positive correlations were found between the changes in HDL-C, Apo AI, and CoQ10 during the follow-up (Table 3).

## 3. Discussion

To the best of our knowledge, this is the first study to demonstrate that among statin-naïve patients with early-onset STEMI, pPCI followed by guideline-directed medical treatment including high-intensity statin therapy is associated with a greatly significant 43% reduction in CoQ10 levels three months after the acute event. The main finding of our investigation from a clinical perspective is that this secondary CoQ10 deficiency is not correlated with FABP3 level, a sensitive and specific biomarker of myopathy. This suggests that contrary to our initial hypothesis and common beliefs among physicians, changes in serum CoQ10 levels are not key players in statin-induced myopathy.

Another important finding is that this CoQ10 depletion did not lead to unfavourable changes in other key biomarkers of HF, inflammation, and cardiac muscle damage (FABP3 initially named heart-type fatty acid-binding protein is also a highly specific and sensitive marker for myocardium injury). Concentrations of NT-proBNP, CRP, IL-6, and FABP3 decreased by 76%, 70%, 88%, and 79%, respectively, and the change in these parameters did not correlate with the change in CoQ10 levels during a 3-month-long statin treatment. Altogether, our results suggest that CoQ10 depletion does not lead to clinically significant consequences, at least in the short term after STEMI.

Controversial results are available in the literature about the potential role CoQ10, or its supplementation, plays in myocardial ischemia or infarction. The heterogeneous results do not lead to meaningful conclusions. Limitations include small sample sizes, lack of any plasma or tissue CoQ10 measurements, missing control CoQ10 values, performing multiple interventions at the same time, and not reporting the intensity of statin treatment known to influence CoQ10 levels [14,16,21,22,23,24]. Reviewing animal experiments is beyond the scope of our research, but it should be noted that several early reports showed that CoQ10 treatment could ameliorate the consequences of myocardial ischemia in experimental animals [11,12,13,25,26]. However, the results of strictly controlled animal experiments can be extrapolated to human studies and clinical practice with great caution only.

STEMI patients receive strict guideline-directed medical treatment including a high-intensity potent statin, beta-blocker (BB), angiotensin-converting enzyme inhibitor (ACE-I), or angiotensin receptor blocker (ARB) and dual antiplatelet treatment (DAPT) unless contraindicated [8]. In addition, selected patients with very high LDL-C levels (>3.4 mmol/L) or familiar hypercholesterolemia can be administered upfront combined lipid-lowering medications including ezetimibe or a PCSK9 inhibitor [10,27]. Medical treatment was administered according to current guidelines in our study as shown in the results. These drugs have the potential to influence the energy metabolism of the myocardium, the pro- and anti-inflammatory pathways, oxidative stress, and CoQ10 levels. For example, it is well known that ACE-I/ARB and BB improve outcomes after ACS by decreasing afterload and myocardial oxygen demand, by acting against ventricular remodelling, or by reducing neurohormonal activation of the sympathetic nervous system. Both classes of agents prevent the derangement of cardiac energy metabolism [28]. Another example is aspirin, a key component of DAPT, which is well known for its complex anti-inflammatory and indirect antioxidant effects [29]. Finally, but most importantly, there is a substantial amount of evidence that statin use is inversely correlated with CoQ10 levels [30]. To the best of our knowledge, only one clinical trial explored the change in CoQ10 levels compared to baseline after STEMI, and 85% of the patients in that study were on statin treatment at enrolment [31]. The authors reported a 34% decrease in mean plasma CoQ10 levels 1 month after STEMI and claimed that patients who had higher plasma CoQ10 concentrations at that timepoint had better left ventricular function at the 6-month follow-up.

Based on the overwhelming amount of evidence supporting the role of LDL-C in the progression of atherosclerosis, the previous LDL-C target values have been lowered by the 2019 European Society of Cardiology/European Atherosclerosis Society (ESC/EAS) lipid guidelines [9]. Accordingly, in patients with a history of ACS, the target LDL-C levels to be achieved are 1.4 mmol/L (55 mg/dL) with at least a 50% reduction in LDL-C. This requires high-intensity statin treatment as basic LLT in virtually all patients. The European DaVinci (EU-Wide Cross-Sectional Observational Study of Lipid-Modifying Therapy Use in Secondary and Primary Care) study assessed the quality of LLT in 18 countries, and the LDL-C target value of 1.4 mmol/L (55 mg/dL) was achieved by less than one-fifth of patients [32]. The situation is worse in Central and Eastern European countries, where target values are achieved at least 10% less frequently when compared to Western and Northern European data [33]. Thus, unfortunately, only a small proportion of patients can reach their individual target LDL-C values mainly due to poor patient adherence and physician inertia [34]. Our findings support these data since the mean LDL-C levels markedly reduced 3 months after STEMI, but they did not reach the target LDL-C values. Often, unfounded fears of side effects contribute to poor therapeutic adherence and persistence. A common belief is that the decreased CoQ10 levels associated with statin treatment might lead to unfavourable energy metabolism in skeletal and cardiac myocytes leading to statin-associated myopathy symptoms (SAMSs) or aggravated HF and that the lower CoQ10 can cause increased inflammation and oxidative stress contributing to atherosclerosis progression or adverse ventricular remodelling. Indeed, it is very important to scientifically address these issues and dispel any misconceptions, since these highly vulnerable post-ACS patients clearly benefit from robust, lifelong LLT. From another point of view, however, if the decrease in CoQ10 levels does indeed have a detectable adverse effect, then eliminating it with CoQ10 supplementation as an add-on therapy might further increase the benefits of statin use.

Reports started to emerge in the early 1990s showing that statin treatment was associated with a significant reduction in CoQ10 levels (e.g., 29% reduction with 80 mg/day lovastatin and 20% reduction with 40 mg/day pravastatin) in a dose-dependent manner in both animal studies and human experiments recruiting healthy volunteers or patients with hypercholesterolemia [35]. This decrease in CoQ10 parallelled the decline of cholesterol and it was eventually detected with every statin tested [30,36]. The reduction in plasma CoQ10 was proportionally greater than the reduction in LDL-C in some studies [37], while it was similar in others [18]. The relative reduction in CoQ10 was less (−43%) than that of LDL-C (−47%) in our study; therefore, due to the marked LDL-C reduction, the CoQ10/LDL-C ratio significantly increased. This is important because CoQ10 is carried in the plasma by lipoproteins with the LDL fraction carrying about 60% of the total serum CoQ10, where it serves to diminish the oxidation of LDL-C [38,39]. Thus, it remains to be decided whether the statin-associated CoQ10 reduction is only an indirect consequence of LDL-C reduction or directly linked to inhibition of the mevalonate pathway. Animal studies proved that depletion occurs in blood and tissues alike, and CoQ10 deficiency is associated with adverse events, like skeletal muscle injury or cardiomyopathy. Importantly, it was also shown that CoQ10 supplementation could completely prevent CoQ10 depletion. In humans, the statin-induced CoQ10 depletion is thought to be well tolerated in younger, healthy patients, especially in the short term, but in patients with preexisting heart disease or acute cardiac injury, there is reasonable evidence to support a detrimental effect [36]. Our patients were relatively young (48.7 ± 6.0 years of age) and this must be taken into consideration when interpreting our results. It must be emphasized though that there are simply no efficacy data to support the routine use of CoQ10 supplementation for preventing statin-related adverse events. Meta-analyses of available randomized trials found that CoQ10 administration could not offer significant benefit in improving SAMS [40,41] and a prospective case-–control study from the placebo-controlled LIPID trial concluded that the CoQ10 reduction experienced after pravastatin administration could not predict the risk of recurrent cardiovascular events [19].

Our study supports current guidelines and provides additional information that high-intensity statin treatment is safe in the STEMI population, where the most effective LLT is required, while the risk of HF, cardiomyopathy, and SAMS is the greatest. Inflammation or oxidative stress are additional concerns in this population because these can promote the progression of atherosclerosis or increase the risk of recurrent vascular events.

FABP3 is an important biomarker for cardiac or skeletal muscle injury. Its physiological role is central in the regulation of energy and lipid metabolism by serving as a lipid chaperone [42]. FABP3 levels are elevated in toxic or drug-induced skeletal muscle injuries [43] in all forms of HF, including HF with preserved ejection fraction [44], in various cardiomyopathies [45], and in acute coronary syndrome [46], and it is a useful marker for ongoing myocardial damage [47]. Our results are in line with these findings. Interestingly, its high sensitivity for striated muscle damage is illustrated by its evolving role as an accepted novel biomarker for peripheral arterial disease, where repetitive tissue ischemia and reperfusion lead to an acquired myopathy with FABP3 elevation [48]. Our results show that FABP3 levels significantly decreased and reached values reported by healthy controls [44] or that of stable diabetic patients with normal ejection fraction [49], which is reassuring and suggests that CoQ10 depletion did not lead to significant myopathy. This was further confirmed by the normal CK values, measured in our cohort as an alternative, although less sensitive, biomarker of muscle injury.

It is also encouraging that CRP and IL-6 values were normal at 3 months after STEMI, again documenting that plasma CoQ10 reduction was not associated with increased levels of inflammation. This is a very important finding because it is well known that in patients with known cardiovascular disease, elevated CRP results in substantially higher rates of recurrent vascular events despite reaching LDL-C goals, a phenomenon called residual inflammatory risk [50]. Several landmark randomized trials showed that reducing this inflammatory risk by anti-inflammatory therapies such as IL-1ß antagonist canakinumab or colchicine on top of standard-of-care medical management could improve outcomes further [51]. Furthermore, a recent major study showed that high-sensitivity CRP was an independent predictor of future cardiovascular events among healthy women [52]. CoQ10 is provided with anti-inflammatory properties [4]. By binding to peroxisome proliferator-activated receptor gamma (PPAR-γ), the CoQ10-PPAR-γ complex can inhibit nuclear factor kappa B (NFκB) leading to the downregulation of an array of proinflammatory cytokine gene expressions, such as IL-6 or tumour necrosis factor alpha. Anti-inflammatory and antioxidant effects are closely related. Thus, it was worrying that CoQ10 depletion could lead to a flare-up of inflammation, but this hypothesis was contradicted by our findings. Median CRP and IL-6 values at three months were 1.8 mg/dL and 2.0 pg/mL, respectively; both values are considered to represent moderate long-term residual inflammatory risk [53], below the inclusion cutoff set forth in the targeted anti-inflammatory trials.

Finally, the relationship between CoQ10 levels and HF outcomes is another intriguing question with a well-established theoretical background, but with very limited high-quality outcome data supporting an actual interaction. Generally, the trials focused on CoQ10 supplementation in HF with various prescription formulas or doses, had small sample sizes, were subject to significant bias, and had conflicting results, which included no effect as well. The detailed discussion of CoQ10 in HF is outside the focus of our study, but few remarks should be noted. A systematic review and meta-analysis of randomized trials found that studies on CoQ10 were significantly biased, leading to uncertain evidence regarding its effectiveness. Compared to a placebo, CoQ10 showed an unclear benefit for reducing mortality (relative risk 0.68, 95% CI 0.45–1.03) and had either minimal or ambiguous effects on HF symptoms [54]. Post-MI patients with ischemic HF form a special subpopulation among individuals with myocardial dysfunction, since on top of guideline-dictated HF management, they require LLT. An important and well-designed study evaluated the association between serum CoQ10 levels and outcomes on statin- or placebo-treated ischemic HF patients with reduced ejection fraction from a pre-specified subgroup of the Controlled Rosuvastatin Multinational Study in Heart Failure (CORONA) trial [55]. They made three relevant observations. First, older, frail patients with more severe HF and multiple comorbidities had lower CoQ10 levels. Second, the intermediate dose of rosuvastatin (10 mg) administered on the active arm was associated with a median 41.9% reduction in plasma CoQ10 levels. Third, after adjusting for multiple variables, the CoQ10 level was not an independent predictor of all-cause mortality or any other composite outcomes, including HF hospitalizations. Although our study was not aimed at investigating outcomes after STEMI, the finding that NT-proBNP levels decreased significantly to a median of 252 pg/mL, well below the rule-in cutoff values for manifest HF (>600 pg/mL), is comforting and argues against significant HF in our early STEMI population despite the CoQ10 depletion. Manifest HF was also not observed during the 3-month follow-up.

Some limitations of our clinical study must be mentioned. Although the statistical power of the study is convincing, enrollment of a larger patient population including elderly subjects, more female patients, subjects with proven statin-associated myopathy, and patients with type 1 and type 2 diabetes mellitus and chronic renal failure could confirm our findings. However, for this trial, we wanted to avoid the confounding effect of comorbidities on CoQ10 levels intentionally; thus, we chose premature STEMI patients as an ideal model population for investigating this topic. Enrolling such patients significantly limited the sample size, since only 16% of the original IMACS populations and 3.2% of our total MI population could be enrolled in the CoQ10 sub-study based on the exclusion and inclusion criteria. The proportion of female patients was only 36%, but this was in line with the accepted epidemiological data on gender differences in the early-onset STEMI population. Determination of intramuscular CoQ10 concentration could provide more data, but invasive procedures such as skeletal muscle or myocardial biopsy are not available in everyday clinical practice. Indeed, in clinical studies, plasma CoQ10 concentrations are widely used for the estimation of CoQ10 status in humans primarily because of the ease of sample collection. Furthermore, the direct association of CoQ10 with lipoprotein particles was not measured, since the determination is not widely available, and since it is a time-consuming and costly process. Another limitation is that only a proportion of our patients (28.3%) reached the desired LDL-C target of <1.4 mmol/L at the 3-month control visit when the second CoQ10 sampling was performed. LLT was titrated further at this timepoint as illustrated in Appendix A. Thus, an even greater decrease in CoQ10 levels could be expected for these post-STEMI patients, if further CoQ10 measurements had been performed when LLT was up-titrated to the desired level. The slow dose escalation is caused by several factors related to suboptimal LLT treatment patterns in Hungary [56]. The main reason is that ezetimibe or PCSK-9 inhibitors are only reimbursed if LDL-C is above 1.8 mmol/L after a three-month-long high-intensity statin treatment or a one-month-long combined statin and ezetimibe therapy, respectively. It would have been interesting to examine the effect of CoQ10 supplementation on LDL-C levels; however, our study was not designed with this in mind. A meta-analysis of randomized trials concluded that CoQ10 supplementation decreased the TC, LDL-C, and TG levels, and increased HDL-C levels [57]. Further interventional trials should clarify whether this could contribute to reaching LDL-C targets among patients after ACS. Finally, measurement of functional markers of muscle damage including electromyography and MRI could affirm the conclusions, but since no SAMSs were reported among our patients, these clinical examinations were not indicated. Our results highlight the importance of human studies which can contribute to clarifying the exact clinical indications of CoQ10 supplementation.

## 4. Materials and Methods

### 4.1. Patient Population

The study design is demonstrated as a flowchart in Appendix A. Investigation of Intestinal Metagenome and cardiovascular (CV) risk factors in patients with Acute Coronary Syndrome (IMACS) is a prospective observational study of non-conventional CV risk factors among patients with STEMI, NSTEMI (type 1 MI), or sudden cardiac death (SCD) caused by myocardial infarction (type 3 MI). All patients were treated at the cardiac intensive care unit of a tertiary cardiovascular centre in Miskolc, Hungary. From 1 January 2021 to 31 December 2022, adult patients above 18 years presenting with acute myocardial infarction were pre-screened (*n* = 2067) and, if eligible, were enrolled (*n* = 410) into the study within 48 h after the onset of ACS symptoms, with a culprit epicardial coronary lesion confirmed by coronary angiography performed during pPCI. The main exclusion criteria were restrictive diets (vegan, vegetarian, gluten-free), pre- or probiotic treatments or food supplements, antibiotic therapy in the past 3 months, excessive alcohol consumption, drug abuse, long-term treatment with proton-pump inhibitors (PPIs), previous MI, myocardial infarction with non-obstructive coronary arteries (MINOCA), known cardiomyopathy, active malignancies, immunosuppression, and significant infectious (hepatitis B and C; tuberculosis; human immunodeficiency virus infection), autoimmune, endocrine, psychiatric, liver (Chid-Pugh B or C stage liver cirrhosis or hepatic failure), gastroenterological (inflammatory bowel disease, malabsorption, malnutrition, irritable bowel syndrome), or renal diseases (GFR ≤ 30 mL/min/1.73 m^2^).

The CoQ10 sub-study included patients with premature STEMI (males < 55, females < 60 years of age) from the original IMACS population, who had no type 1 or type 2 diabetes mellitus, had no significant chronic kidney disease (GFR ≥ 60 mL/min/1.73 m^2^), and were not receiving lipid-lowering therapy prior to the acute event (*n* = 67).

STEMI was defined based on the fourth universal definition of MI [58] set forth by the European Society of Cardiology (ESC) using the following criteria: (1) symptoms of myocardial ischemia, (2) newly occurring ST elevation at the J-point in at least two contiguous leads with a magnitude of ≥2 mm for men and ≥1.5 mm for women in leads V2-V3 or ≥1 mm in any other contiguous precordial or limb leads for both sexes on a 12-lead electrocardiography (ECG), and (3) identification of a coronary thrombus by angiography. Rises and/or falls of cardiac troponin T (cTnT) values with at least one value above the 99th percentile of the upper reference limit were confirmed in all patients.

All participants provided written informed consent, and the study was carried out in accordance with the Declaration of Helsinki. The study protocol was approved by the Local and Regional Ethical Committees (RKEB/IKEB: 4739/2017, date of approval: 20 February 2017, and ETT/TUKEB: 7324-9/2017/EÜIG, date of approval: 22 April 2020, respectively).

### 4.2. Patient Procedures and Assessments

Patients were pretreated with dual antiplatelet therapy (DAPT) by the emergency medical services using 600 mg clopidogrel and 300 mg aspirin before the urgent pPCI. Eligible patients were switched to a potent, 3rd generation P2Y_12_/adenosine diphosphate (ADP) receptor inhibitor prasugrel or ticagrelor according to international guidelines and national reimbursement regulations. A coronary intervention was carried out through the radial artery with a door-to-balloon time of <60 min. Drug-eluting stents (DESs) were used for revascularisation when this was applicable. After PCI, patients received guideline-directed standard of care using DAPT, beta-blockers, angiotensin-converting enzyme inhibitors (ACE-Is) or angiotensin receptor blockers (ARBs), high-intensity statin treatment with at least 20 mg rosuvastatin or 40 mg atorvastatin, and proton-pump inhibitor (PPI) for ulcer prevention unless contraindicated. In particular, the long-term target LDL-C level was defined as <1.4 mmol/L (<55 mg/dL) with at least a 50% reduction in LDL-C. Patients participated in lifestyle counselling and cardiac rehabilitation.

Demographic parameters, comorbidities, cardiovascular risk factors including smoking status, baseline and three-month routine laboratory values, Killip classification of HF, Acute Physiology and Chronic Health Evaluation II (APACHE-II) score, Global Registry of Acute Coronary Events (GRACE) score, and concomitant medications were assessed. Transthoracic echocardiography was performed at admission. Procedural details related to pPCI, such as the extent of atherosclerotic coronary involvement (single or multivessel), culprit vessel, type of coronary intervention (stent implantation, balloon angioplasty, intention for coronary artery bypass graft surgery), and pre- and post-PCI Thrombolysis in Myocardial Infarction (TIMI) flow were recorded. Statin intensity was defined based on the American College of Cardiology and American Heart Association (ACC/AHA) classification of statin dosing and intensity [59]. Briefly, the high-intensity statin regimens included rosuvastatin 20–40 mg and atorvastatin 40–80 mg, while the moderate-intensity regimens included atorvastatin 10–20 mg and rosuvastatin below 20 mg (or, theoretically, simvastatin 20–40 mg and fluvastatin 80 mg should a switch in the choice of statin occur during follow-up).

Venous serum and plasma samples were put into Vacutainer^®^ tubes, taken from each patient at the admission and after 3-month follow-up. Samples were separated at 3500 g 10 min +4 °C. For the subsequent analyses, 0.5 mL aliquots of sera and plasma specimens were stored at −80 °C.

A long-term remote follow-up visit was performed for each accessible patient between 7 December 2024 and 12 December 2024 to assess potential late-onset symptoms of statin-induced myopathy using the Statin-Associated Muscle Symptom Clinical Index (SAMS-CI) questionnaire [60].

### 4.3. Measurement of Routine Laboratory Parameters

The routine laboratory parameters were determined from frozen sera at the Department of Laboratory Medicine, Faculty of Medicine, University of Debrecen, Hungary. Triglyceride and total cholesterol levels were measured by enzymatic, colorimetric tests by a Roche Cobas^®^ 600 analyzer (Roche Diagnostics, Mannheim, Germany). HDL-C and LDL-C levels were determined by homogenous, enzymatic, and colorimetric assays (Roche HDL-C plus 4th generation and Roche LDL-C plus 3rd generation, respectively). Apolipoprotein AI (Apo AI) and apolipoprotein B100 (Apo B100) examinations were performed by immunoturbidimetric assays (Tina-quant apolipoprotein A-I ver. 2 and Tina-quant apolipoprotein B ver. 2, respectively). C-reactive protein (CRP) and interleukin-6 (IL-6) levels were determined via electro-chemiluminescent immunoassays on a Cobas^®^ e411 analyzer (Roche Diagnostics). N-terminal fragment pro-brain natriuretic peptide (NT-proBNP), aspartate aminotransferase (AST), alanine transferase (ALT), creatine kinase, gamma-glutamyl transpeptidase (γ-GTP), creatinine, and uric acid were measured via kinetic colorimetric assays on a Cobas^®^ 8000 analyzer (Roche Diagnostics).

### 4.4. Determination of Plasma CoQ10 Concentration

The plasma CoQ10 concentration was determined using ultra-high-performance liquid chromatography–tandem mass spectrometry (UHPLC/MSMS) at Proteomics Core Facility, Department of Biochemistry and Molecular Biology, Faculty of Medicine, University of Debrecen, Hungary. Coenzyme Q10, Coenzyme Q10-D9 (isotopic purity ≥ 98 atom% D_9_), 2-propanol, formic acid, and hexane were purchased from Sigma-Aldrich (St. Louis, MO, USA). Methanol, acetonitrile, ethanol, and HPLC-grade water were obtained from VWR Ltd. (Radnor, PA, USA).

For the preparation of the calibration standard solution, the stable isotope-labelled Coenzyme Q10-D9 was used as an internal standard. The standard solution of Coenzyme Q10 and Coenzyme Q10-D9 was prepared in ethanol in 500 µg/mL final concentration and stored at −20 °C until utilization. Before utilization, the thawed aliquots were sonicated for 15 min in a water bath. Working solutions with 5 µg/mL Coenzyme Q10 and 10 µg/mL Coenzyme Q10-D9 were prepared. The calibration standards were prepared from the working solution through serial dilutions with ethanol–water (8:2) in the range of 5 ng/mL–1250 ng/mL and spiked with 1 µL of 10 µg/mL Coenzyme Q10-D9 solution.

For the preparation of plasma samples, 250 µL of plasma, 10 µL of 10 µg/mL Coenzyme Q10-D9 solution, 400 µL 100% methanol, and 1000 µL 100% hexane were added and rotated in a rotary shaker for 15 min (700 rpm). After precipitation, samples were centrifuged at room temperature (5 min, 6100 rcf); the supernatant was transferred to a new tube, dried in a Concentrator (Eppendorf), and redissolved in 200 µL 100% acetonitrile. The supernatant was transferred to an amber glass chromatography vial.

A total of 5 µL of sample or calibration standard solution was injected into the UPLC. Liquid chromatographic separation was performed on an Acquity H-class UPLC system (Waters, Milford, MA, USA) controlled by the Empower 3 software (Waters, Milford, MA, USA). The separation of Coenzyme Q10 was carried out on an AccQ-tag Ultra C18 column (1.7 µm; 2.1 × 100 mm, Waters, Milford, MA, USA) guarded by an Acquity in-line filter (0.2 µm; 2.1 mm, Waters, Milford, MA, USA). A 10 min isocratic elution mode was used with a 0.40 mL/min flow rate and 40 °C column temperature. The mobile phase consisted of 90% acetonitrile, 10% 2-propanol, and 0.1% formic acid.

SRM-based targeted mass spectrometry analyses were carried out on a 5500QTRAP mass spectrometer (Sciex, Framingham, MA, USA) controlled by the Analyst software (version 1.6.3, Sciex, Framingham, MA, USA). Electrospray ionization with 5500 V spray voltage was used and the positive ion mode SRM spectra were recorded. Other acquisition parameters were as follows: the ion source gas 1 was set to 30 psi; the ion source gas 2 was set to 50 psi; the curtain gas was 30 psi, and the source temperature was 500 °C. The detailed parameters of the SRM experiment are the following: Q1/Q3 = 863.6/197.2 *m*/*z* of Coenzyme Q10, Q1/Q3 = 872.8/206.2 *m*/*z* of Coenzyme Q10-D9, retention time 5.99 min, declustering potential 60 eV, and collision energy 45 eV and 75 eV, respectively. The SRM spectra were analyzed with the Skyline software version 23.1.0. [61].

### 4.5. Determination of Oxidized Low-Density Lipoprotein (oxLDL) Levels

The oxLDL levels of the serum were measured by a sandwich ELISA method (Mercodia AB, Uppsala, Sweden). The coefficient variations of the intra-assay were from 5.5% to 7.3% and the inter-assay were from 4% to 6.2%. The sensitivity was 1 mU/L and values were expressed as U/L.

### 4.6. Fatty Acid-Binding Protein 3 (FABP3) Measurement

The serum concentration of FABP3 was detected from undiluted samples using a commercially available duoset ELISA kit (R&D Systems Europe Ltd., Abington, UK). The values were expressed as ng/mL.

### 4.7. Statistical Analyses

Statistical analyses were performed using Statistica 13.5.0.17 software (TIBCO Software Inc., Palo Alto, CA, USA). Graphs were made using GraphPad Prism 6.01 (GraphPad Prism Software Inc., Boston, MA, USA). Before the study, sample size calculation was performed with the SPH Analytics online calculator (SPH Analytics Ltd., Alpharetta, GA, USA) to validate the difference in plasma CoQ10 levels during the follow-up. The required sample size was *n* = 39 with 0.8 power, while the actual statistical power was 0.9607 for 67 patients. A Kolmogorov–Smirnov test was used for testing the normality of variables and values were presented as mean ± standard deviation (SD) or median (interquartile ranges). During the follow-up, data comparisons of patients with AMI were performed by repeated measures of the analysis of variance (ANOVA) or by Wilcoxon matched paired test using Bonferroni correction. In the case of multiple comparisons, we labelled only *p*-values which passed the Bonferroni threshold. Correlations between continuous variables were assessed by a linear regression analysis using Pearson’s test. Multiple regression analysis (backward stepwise method) was performed to determine the significant predictors of CoQ10 levels. The results were considered to be significant at *p* < 0.05.

## 5. Conclusions

Our data demonstrate that statin treatment after STEMI is associated with significantly decreased plasma CoQ10 level, which might be the consequence of the decrease in the serum levels of Apo B100 containing lipoproteins, but inhibition of the mevalonate pathway can also play a contributory role. However, the lack of correlation between changes in FABP3 and CoQ10 levels indicates that the drop in plasma CoQ10 level may not be a key player in the pathomechanism of statin-induced muscle damage. Therefore, statin treatment is recommended for all STEMI patients according to the current guidelines. If the reduction in CoQ10 is a concern, then the potential of alternative lipid-lowering therapies (e.g., ezetimibe, PCSK9 inhibitors) should be exploited, because these agents do not affect endogenous CoQ10 synthesis according to our current knowledge. The results indicate that routine measurement of plasma CoQ10 may not be suitable to evaluate the need for CoQ10 supplementation to reduce the risk of statin-induced muscle damage. Clinical investigations specifically designed to address the effect size of statin treatment on CoQ10 biosynthesis through the mevalonate pathway are needed. In addition, randomized studies on larger patient populations, with accepted cardiovascular outcomes as the primary endpoint, are of major interest to determine the further possible consequences of decreased CoQ10 levels after statin treatment and identify patient populations that can benefit from CoQ10 supplementation after STEMI.

## Figures and Tables

**Figure 1 ijms-26-00106-f001:**
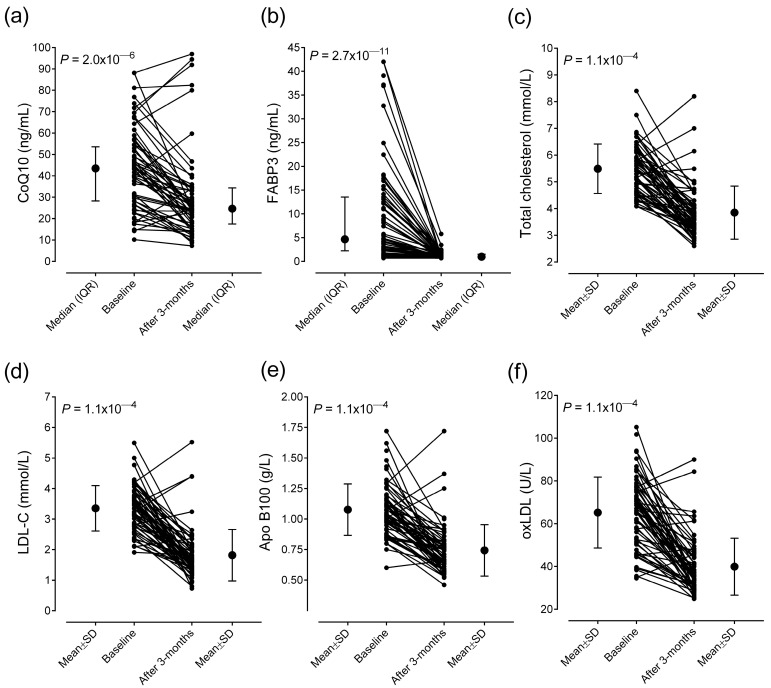
Median (**a**,**b**) or mean (**c**–**f**) and individual levels of plasma CoQ10 (**a**) and serum levels of FABP3 (**b**), total cholesterol (**c**), LDL-C (**d**), Apo B100 (**e**), and oxLDL (**f**) in patients with STEMI (*n* = 67) at admission (baseline) and after the 3-month statin therapy. Abbreviations: Apo B100, apolipoprotein B100; CoQ10, Coenzyme Q10; FABP3, fatty acid-binding protein-3; LDL-C, low-density lipoprotein cholesterol; oxLDL, oxidized LDL.

**Figure 2 ijms-26-00106-f002:**
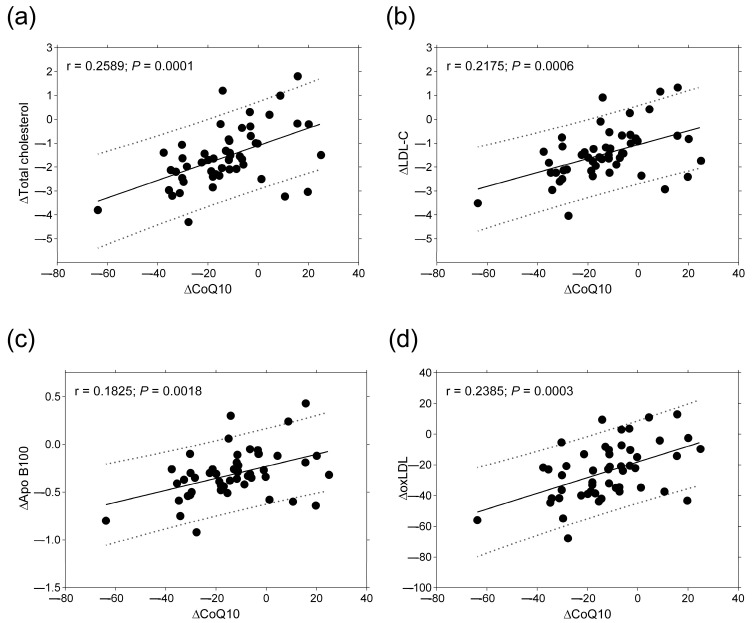
Correlations between the change in serum levels of total cholesterol (**a**), LDL-C (**b**), Apo B100 (**c**), and oxLDL (**d**) and the change in plasma CoQ10 level in patients with STEMI (*n* = 67) during the 3-month statin therapy. Abbreviations: Apo B100, apolipoprotein B100; CoQ10, Coenzyme Q10; LDL-C, low-density lipoprotein cholesterol; oxLDL, oxidized LDL.

**Table 1 ijms-26-00106-t001:** Characteristic of enrolled patients with ST-elevation myocardial infarction (STEMI).

	Patients with STEMI
Number of patients	67
Males (*n*; %)	43; 64.2
Age—overall (years)	48.7 ± 6.0
Body mass index (kg/m^2^)	28.1 ± 5.3
GRACE score	82.9 ± 15.6
APACHE-II score	5.6 ± 2.3
**Cardiovascular risk factors**	
Smoking (*n*; %)	54; 80.6
Hypertension (*n*; %)	43; 64.2
Diabetes (*n*; %)	0; 0.0
Peripheral artery disease (*n*; %)	3; 4.5
Coronary artery disease (*n*; %)	2; 3.0
Stroke/TIA (*n*; %)	2; 3.0
**Culprit lesion**	
Single vessel (*n*; %)	28; 41.8
Multivessel (*n*; %)	39; 58.2
Left ventricular ejection fraction (%)	51.3 ± 13.0
LM (*n*; %)	1; 1.5
LAD (*n*; %)	31; 46.3
CX (*n*; %)	8; 11.9
RCA (*n*; %)	25; 37.3
**Medications at admission**	
Statin (*n*; %)	0; 0.0
Ezetimibe (*n*; %)	0; 0.0
Fibrate (*n*; %)	0; 0.0
PCSK9 inhibitors (*n*; %)	0; 0.0
Acetylsalicylic acid (*n*; %)	8; 11.9
ACE-I/ARB (n, %)	27; 40.3
BB (n, %)	18; 26.9
CCB (n, %)	21; 31.3
**Medications at 3-month follow-up**	
Statin (*n*; %)	66; 98.5
Ezetimibe (*n*; %)	2; 3.0
Fibrate (*n*; %)	2; 3.0
PCSK9 inhibitors (*n*; %)	0; 0.0
Acetylsalicylic acid (*n*; %)	64; 95.5
ACE-I/ARB (*n*; %)	59; 88.1
BB (*n*; %)	61; 91.0
CCB (*n*; %)	9; 13.4

Abbreviations: ACE-Is/ARBs, angiotensin-converting enzyme inhibitors/angiotensin II receptor blockers; APACHE II, acute physiology and chronic health evaluation score; BBs, beta-blockers; CCBs, calcium channels blockers; CX, circumflex artery; GRACE, global registry of acute coronary events risk score; LAD, left anterior descending coronary artery; LM, left main coronary artery; PCSK9, proprotein convertase subtilisin/kexin type 9; RCA, right coronary artery; TIA: transient ischaemic attack. Continuous variables are presented as mean ± SD, while categorical variables are expressed as numbers (portion).

**Table 2 ijms-26-00106-t002:** Laboratory parameters of patients with STEMI before and after 3-month statin treatment.

Variable	At Admission	After 3-Month Follow-Up	*p*-Value
CoQ10 (ng/mL)	43.29 (28.16–54.57)	24.66 (17.8–34.4)	2.0 × 10^−6^
FABP3 (ng/mL)	4.66 (2.21–13.53)	0.98 (0.83–1.46)	2.7 × 10^−11^
NT-proBNP (ng/L)	1058 (392–2128)	252 (125–745)	5.7 × 10^−8^
Creatine kinase (U/L)	1061 (283–2080)	59 (37–90)	1.6 × 10^−11^
CRP (mg/L)	5.9 (2.7–9.7)	1.8 (0.8–3.4)	1.1 × 10^−7^
IL-6 (pg/mL)	16.9 (7.7–31.5)	2.0 (1.5–4.5)	2.1 × 10^−10^
Triglyceride (mmol/L)	1.9 (1.4–2.7)	1.5 (1.1–1.9)	4.1 × 10^−6^
Total cholesterol (mmol/L)	5.5 ± 0.9	3.8 ± 1.0	1.1 × 10^−4^
CoQ10/total cholesterol ratio	8.06 ± 3.57	7.94 ± 5.01	n.s.
HDL-C (mmol/L)	1.0 ± 0.3	1.1 ± 0.2	n.s.
Apo AI (g/L)	1.30 ± 0.29	1.42 ± 0.27	3.3 × 10^−4^
LDL-C (mmol/L)	3.4 ± 0.8	1.8 ± 0.8	1.1 × 10^−4^
CoQ10/LDL-C ratio	13.50 ± 5.94	17.69 ± 11.36	2.1 × 10^−3^
Apo B100 (g/L)	1.08 ± 0.21	0.74 ± 0.21	1.1 × 10^−4^
CoQ10/Apo B100 ratio	41.92 ± 19.03	42.23 ± 28.76	n.s.
Oxidized LDL (U/L)	65.2 ± 16.6	39.9 ± 0.8	1.1 × 10^−4^
AST (U/L)	138 (62–231)	22 (17–27)	5.8 × 10^−11^
ALT (U/L)	20 (12–36)	13 (8–21)	7.2 × 10^−5^
γ-GTP (U/L)	30 (19–46)	29 (19–46)	n.s.
Creatinine (µmol/L)	76.2 ± 29.9	80.0 ± 35.8	n.s.
Uric acid (µmol/L)	5.3 ± 2.0	5.9 ± 2.0	2.7 × 10^−3^

Abbreviations: ALT, alanine transaminase; Apo AI, apolipoprotein AI; Apo B100, apolipoprotein B100; AST, aspartate aminotransferase; CoQ10, Coenzyme Q10; CRP, C-reactive protein; FABP3, fatty acid-binding protein-3; γ-GTP, gamma-glutamyl transpeptidase; HDL-C, high-density lipoprotein cholesterol; IL-6, interleukin-6; LDL, low-density lipoprotein; LDL-C, low-density lipoprotein cholesterol; NT-proBNP, N-terminal fragment pro-brain natriuretic peptide; n.s. non-significant. Data are presented as mean ± SD or median (interquartile ranges) and the statistical differences were calculated by repeated measures ANOVA (Tukey test) or Wilcoxon matched paired test after using Bonferroni correction (*p* < 2.9 × 10^−3^).

**Table 3 ijms-26-00106-t003:** Correlations of the change in plasma CoQ10 levels with the change in laboratory parameters at admission and after 3-month statin treatment in STEMI patients.

	∆CoQ10 (ng/mL)	
Variable	r	*p*-Value
∆FABP3 (ng/mL)	0.116	0.245
∆NT-proBNP (ng/L)	−0.215	0.131
∆Creatine kinase (U/L)	0.156	0.281
∆CRP (mg/L)	−0.222	0.117
∆IL-6 (pg/mL)	−0.063	0.657
∆AST (U/L)	0.078	0.589
∆ALT (U/L)	0.154	0.286
∆γ-GTP (U/L)	0.109	0.453
∆Creatinine (µmol/L)	0.032	0.826
∆Uric acid (µmol/L)	−0.004	0.978
∆Triglyceride (mmol/L)	0.212	0.136
∆HDL-C (mmol/L)	0.306	0.029
∆Apo AI (g/L)	0.301	0.030

Abbreviations: ALT, alanine transaminase; Apo AI, apolipoprotein AI; AST, aspartate aminotransferase; CoQ10, Coenzyme Q10; CRP, C-reactive protein; ∆, change after 3-month statin treatment compared to baseline; FABP3, fatty acid-binding protein-3; γ-GTP, gamma-glutamyl transpeptidase; HDL-C, high-density lipoprotein cholesterol; IL-6, interleukin-6; NT-proBNP, N-terminal fragment pro-brain natriuretic peptide; STEMI, ST-elevation myocardial infarction.

## Data Availability

All data generated or analyzed during the current study are available from the corresponding author upon reasonable request.

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
