# Peer review of "Newly Initiated Statin Treatment Is Associated with Decreased Plasma Coenzyme Q10 Level After Acute ST-Elevation Myocardial Infarction"

_ijms, 2024, doi:10.3390/ijms26010106_

Round 1

Reviewer 1 Report

Comments and Suggestions for Authors

The study was aimed at assessing the effect of newly initiated statin treatment on serum CoQ1028 level after acute ST-elevation myocardial infarction. Results showed that statin treatment after STEMI is associated with significantly decreased plasma CoQ10 level, while the lack of correlation between changes in FABP3 and CoQ10 levels indicated that the drop of plasma CoQ10 level might be not  a pivotal mechanism in the statin-induced muscle damage. The authors concluded that routine measurement of plasma CoQ10 may not be appropriate for evaluating the need for CoQ10 supplementation to minimize the risk of statin-induced muscle damage. Subject matter is of some interest, however several issues should be better addressed.

General comments

The routine evaluation of CoQ10 plasma level as marker for supplementation is uncommon in the clinical setting manly due to the multifactorial pathogenesis of statin-induced muscle damage, encompassing gene regulation and polymorphisms, mitochondrial disfunction, HMG-CoA reductase pathway, protein prenylation, Atrogin-1 calcium signaling and glycine amidinotransferase, as well as alteration of immune response. The rationale for recommending statin users oral CoQ10 supplementation is an ongoing discussion among clinicians and there is still a void of scientific evidence supporting beneficial effects of this therapeutical approach on these patient populations. In this frame, a recent clinical study (doi: 10.3390/antiox11091698) has highlighted that long term assumption of coenzyme Q10 supplementation in statin treated  patients did not affect muscle CoQ10 levels or mitochondrial function, and individual changes in muscle CoQ10 levels did not  correlate with individual changes in intensity of myalgia, despite the increase in plasma CoQ10 concentration in the CoQ10 group. Moreover, the use of coenzyme Q10 supplementation is hampered by the unfavorable pharmacokinetics of this vitamin like lipophilic cofactor, owing to its  low bioavailability and its selective absorption by different tissues and intracellular translocation. Notably, despite the large amounts and the critical role of the cofactor in cardiac, skeletal muscle tissues where mitochondria are abundant, they appear particularly refractory to exogenous CoQ10 uptake compared to other tissues such as blood, spleen and liver (doi.org/10.1016/S0891-5849(02)01357-6). These observations have already brought attention to the marginal role played by the CoQ10 on the statin-induced myopathy and to the factors underlying the frequent ineffectiveness of  the supplementation.

Specific comments

Abstract: (line 38) the sentence “the musculoskeletal safety of statins” should be amended since myopathy is a common adverse effect of this class of drugs.

Introduction: Hypothesis and study objective should be clearly stated

Results: although the authors demonstrated the lack of correlation between FABP3 and CoQ10 levels it's important to explain why they didn't consider CPK which represents a reliable and clinically accepted marker to monitoring statin myopathy instead of FABP3.

Discussion: this section should be thoroughly revised as it is redundant and poorly focused on the data. The subsection on animal models should be deleted or summarized.

Reviewer 2 Report

Comments and Suggestions for Authors

see appended document

Round 2

Reviewer 1 Report

Comments and Suggestions for Authors

The manuscript has been improved and deserves to be published